# Chronic Effect of Fatmax Training on Body Weight, Fat Mass, and Cardiorespiratory Fitness in Obese Subjects: A Meta-Analysis of Randomized Clinical Trials

**DOI:** 10.3390/ijerph17217888

**Published:** 2020-10-28

**Authors:** Isaac A. Chávez-Guevara, René Urquidez-Romero, Jorge A. Pérez-León, Everardo González-Rodríguez, Verónica Moreno-Brito, Arnulfo Ramos-Jiménez

**Affiliations:** 1Institute of Biomedical Sciences, Autonomous University of Ciudad Juarez, Ciudad Juárez, Chihuahua 32310, Mexico; isaac.chavez@uacj.mx (I.A.C.-G.); rurquide@uacj.mx (R.U.-R.); alberto.perez@uacj.mx (J.A.P.-L.); 2Faculty of Medicine and Biomedical Sciences, Autonomous University of Chihuahua, Circuito Universitario, Campus II, Chihuahua 31109, Mexico; evegonzal@uach.mx (E.G.-R.); vmoreno@uach.mx (V.M.-B.)

**Keywords:** physical exercise, physical fitness, energy metabolism

## Abstract

Exercise training performed at the maximal fat oxidation intensity (FMT) stands out as a potential treatment of overweight and obesity. This work is a meta-analysis of randomized clinical trials of studies about the effect of FMT on fat mass and maximal oxygen consumption using PubMed, SCOPUS, EBSCOhost, and ScienceDirect as databases. Two independent reviewers selected 11 trials from 356 publications identified by the following keywords: fatmax, lipoxmax, maximal fat oxidation, peak of fat oxidation, physical training, physical exercise, body fat (BF), fat mass, overweight, and obesity. The risk of bias was assessed following the Cochrane Guidelines. The pooled mean difference was computed for each outcome with the random-effects model and the inverse-variance method. The meta-analysis was performed with the RevMan software v 5.3, and the heterogeneity across studies by the I^2^. The statistical significance was accepted at *p* < 0.05. Results showed that the FMT reduced body weight (MD = −4.30 kg, *p* < 0.01, I^2^ = 0%), fat mass (MD = −4.03 kg, *p* < 0.01, I^2^ = 0%), and waist circumference (MD = −3.34 cm, *p* < 0.01). Fat-free mass remains unchanged (MD = 0.08 kg, *p* = 0.85), but maximal oxygen consumption increased (MD = 2.96 mL∙kg^−1^∙min^−1^, *p* < 0.01, I^2^ = 0%). We conclude that FMT at short and medium-term (eight to twenty weeks) reduces body weight and BF, increasing cardiovascular fitness in low physical fitness people with obesity.

## 1. Introduction

Currently, overweight and obesity (O/Ob) are the most significant public health challenges, affecting more than 1.9 billion adults worldwide [1]. Both conditions are characterized by increased adiposity, which leads to lipid storage on ectopic tissues like cardiac and skeletal muscle resulting in several metabolic and cardiovascular disorders like insulin resistance, oxidative stress, and an atherogenic lipid profile, all of which increase cardiovascular risk [2,3]. O/Ob are related to impaired fatty acid oxidation in skeletal muscle [4], which can be improved by physical exercise [5]. Of all physical exercise variants, exercise at the intensity that elicits maximal fat oxidation (fatmax) stands out as a potential treatment for body fat (BF) reduction in individuals with O/Ob, due to the following reasons: (I) fatmax training (FMT) does not represent a muscle and articular injury risk for sedentary people with obesity, as fatmax ranged from 40% to 50% of maximal oxygen consumption (VO_2_max) in this population [6,7], (II) FMT meets the American College of Sports and Medicine guidelines to initiate physical conditioning in individuals with obesity [8], (III) lipid metabolic pathways are maximally activated during fatmax, which might increase lipid oxidation and reduce lipid accumulation in skeletal muscle [9], (IV) FMT allows the highest fat utilization during physical exercise, which might contribute to reaching a negative fat balance. However, studies where fatmax is applied as a form of cardiovascular treatment or to reduce fat mass and body weight are scarce, and the results are inconsistent. In this sense, Venables and Jeukendrup (2008) showed that short FMT intervention (four weeks) without caloric restriction is insufficient to decrease BF or increase VO_2_max [10]. Instead, eight to ten weeks of FMT reduced fat mass by 2 kg, augmented VO_2_max 10–20%, and improved insulin sensitivity, augmented lipid oxidation, and increased muscle oxidative capacity in diabetic and non-diabetic obese individuals [11,12,13,14]. The above suggests that mid-term FMT could be an effective therapy to improve metabolic and physical health in people with O/Ob.

Nevertheless, there is only one meta-analysis on the effect of FMT in physical health. This study reported a decrease in body weight (BW) on individuals with different health conditions: obesity, metabolic syndrome, HIV-infection, and psychiatric diseases [15]. Nevertheless, due to differences in population characteristics and methodological approaches, a high heterogeneity among studies (I^2^ = 85%) was observed, limiting its usefulness. Besides, the work did not study the effect of FMT without caloric restriction on body fat, the principal component related to metabolic disorders. Further, we did not find a meta-analysis on the FMT effect on cardiorespiratory fitness (CRF), an essential component of physical health strongly associated with cardiovascular risk and all-mortality causes [16].

Most studies report a fatmax between 40% to 50% of VO_2_max in obese subjects; however, there are large intra-individual variations, independent of sex (30–65% of VO_2_max) [6,7]. Such variations are related to differences in CRF, body composition, and genotype [17,18], indicating that at moderate exercise intensity, some individuals will be working within the fatmax zone, while others are below or above, resulting in different fat oxidation rates among individuals. For these arguments, moderate exercise training should not be considered as fatmax training, as exercise at maximal fat oxidation (MFO) requires individual testing and prescription.

Due to the above reasons, the present meta-analysis aimed at defining the effect of FMT on body composition and CRF in O/Ob subjects.

## 2. Materials and Methods

### 2.1. Identification of Manuscripts

The search was carried out by two independent researchers (IACG, JHM) using PubMed, SCOPUS, EBSCOhost, and ScienceDirect databases. The following search string was used in all databases: ((“fatmax” OR “lipoxmax” OR “maximal fat oxidation” OR “peak of fat oxidation”) AND (“physical training” OR “physical exercise”) AND (“body fat” OR “fat mass”) AND (“overweight” OR “obesity”)). Because the MFO and fatmax concepts were defined in 2001 [9], the publication date was delimited from 1 January 2001, to 31 August 2020. PRISMA guidelines were used for the report [19] (Figure 1).

### 2.2. Inclusion Criteria

Humans randomized clinical trialsPeer-reviewed articlesPeople with BMI ≥ 25 kg∙m^2^Studies where fatmax was used as a training strategyStudies where MFO and fatmax were measures by indirect calorimetryStudies where BF and CRF were reported as a primary or secondary outcome

### 2.3. Exclusion Criteria

Intervention period lower than eight weeksUnsupervised exercise sessions during trialsPapers that did not specify the physical activities on the applied exercise protocolStudies where men and women were analyzed as a single groupStudies performed on individuals with physical disabilitiesDocuments written in languages other than English and Spanish

### 2.4. Data Extracted

The authors, year, experimental design, sample size, intervention period, population (race, sex, age, BMI, health status and physical activity), diet intervention, and training characteristics (exercise mode, exercise intensity, frequency, and duration) were noted. BW, BF percentage (%BF), fat mass (FM), fat-free mass (FFM), waist circumference (WC), VO_2_max, and MFO were recorded as outcomes. Papers were listed by publication date to provide a chronological order (Table 1).

### 2.5. BMI Classification

For the present review, participants were classified as overweight or obese people using BMI reported in each study. Children and adolescents were considered as obese if their BMI value was above the 95th percentile. For adults, BMI cutoffs-values for each ethnic population were employed if available in the literature. For those ethnic populations without specific BMI, cutoffs-values of the World Health Organization were utilized. 

### 2.6. Risk of Bias

The risk of bias (selection, performance, detection, attrition, and reporting) for each article was assessed by two independent reviewers (IACG and ARJ), following the Cochrane handbook guidelines of systematic reviews [20]. A third researcher independently resolved disagreements (RUR). The Review Manager software (Version 5.3, The Cochrane Collaboration, London, UK) was used for the evaluation of the risk of bias.

### 2.7. Statistical Analysis

To estimate the effect of FMT, mean differences (baseline vs. post-intervention) and mean standard errors were pooled for each outcome. A random-effects model and inverse variance method were used to compute the forest plot. Heterogeneity across studies was estimated using the I^2^ statistic, which indicates the proportion of the variability observed due to the heterogeneity of the studies and not randomly. Statistical analyses were performed with the RevMan software v 5.3., and statistical significance was accepted at *p* < 0.05.

The robustness of BW and FM results was explored by three sensitivity analysis: (I) One by one exclusion of all studies (Figure A1A,B), (II) removal of studies performed in adolescents (Figure A1C), (III) removal of studies performed in male participants (Figure A1D). The last two analyses were performed because the majority of studies were performed on female and adult participants. Besides, studies performed in obese adolescents had a low methodological quality (n ≤ 11; training was performed by playground activities).

Publication bias was assessed by using funnel plots, plotting the standard error against the mean difference (Figure A2 and Figure A3). If asymmetry was visually identified, the trim and fill method [21] was used to estimate effect size modifications. Dissimilarities between the observed and adjusted pooled effect size was explored through a *t*-test for independent means (*p* ≤ 0.05). 

## 3. Results

### 3.1. Characteristics of the Search

According to the above methodology, 356 manuscripts were identified, and 93 duplicates were excluded. Afterward, abstracts of the 263 remaining documents were read, excluding 243 because they were not relevant or did not meet the inclusion criteria (Figure 1). The remaining 20 manuscripts were then entirety read, and five of them were excluded (Table 1) due to the following exclusion criteria:Participants were not randomly assigned to the experimental and control groupsThe authors did not specify exercise modeThe authors grouped men and women into the same groups for statistical analysis

Besides, two papers studied the same individuals, which results in the exclusion of one of them. Another three articles written in French were excluded, leaving a total of 11 papers for synthesis and analysis. 

### 3.2. Risk of Bias

(I) In all studies, participants were randomly assigned to the groups; however, seven did not detail the randomization method [22,23,24,25,27,28,29]. (II) Only one study provides information about the allocation concealment [26]. (III) Only one study reported not blinding protocol to participants and researchers [26]; the others do not report this risk of bias. (IV) One study maintained the blinding of outcome assessment [31], and one reported not having done it [26]; the others do not report this risk of bias. (V) In three studies, there were no dropouts [22,23,24]. The desertion ratio of the participants in the remaining trials ranged from 4% to 20%; these studies specified the abandonment reason [25,26,27,28,29,30,31,32]. (VI) All the studies presented data from the variables proposed in their objectives, and the results were discussed and compared with previous works. (VII) No other bias sources were identified like incomplete methodology, conflicts of interest, or ethical problems. The summary and details of the risk of bias are found in Figure 2.

### 3.3. Subjects Characteristics

Three studies were performed in Chinese adults [25,27,31] and three in Chinese older adults [29,30,32], of which two included patients with type two diabetes [29,32]. One study was performed in French adults [26], three in Tunisian adolescents [22,23,24], and one in Chinese boys [28]. Specific BMI cutoff were available for Chinese but not for French adults. Chinese adult participants were classified as obese according to the BMI criteria for the Chinese population [33]. Likewise, with the BMI criteria of the World Health Organization, French adults must be considered as obese individuals (BMI ≥ 30). In addition, the authors report a BMI value above the 95th percentile for adolescent participants, classifying them as individuals with obesity [34]. 

Regarding sex, seven studies included females [23,25,26,27,29,30,31], while three included male subjects [22,24,28]. One study included male and female participants, but analyzed them into different statistical groups and reported all data for both sexes [32].

Concerning physical activity, eight manuscripts informed that recruited individuals were not enrolled in any exercise program or realize less than 60 min/week of physical exercise before enrollment in the study [25,26,27,28,29,30,31]. However, three studies did not mention the physical activity of participants prior to training intervention [22,23,24].

### 3.4. Study Design

This meta-analysis focuses on the isolated effect of FMT, ruling out any other type of physical training, diet treatment, or caloric restriction. All studies were randomized clinical trials. Three studies compared FMT vs. FMT + hypocaloric diet, and fatmax vs. hypocaloric diet without exercise [22,23,24]. One study compared FMT vs. moderate-intensity training, another one compared FMT vs. high-intensity interval training and resistance training [31]. Six studies compared FMT vs. a control group without exercise nor diet intervention [25,27,28,29,30,32]. 

### 3.5. Interventions Characteristics

The training volume was 120–360 min per week, and the intervention period ranged from 8 to 20 weeks. Three different physical activities were identified: playground activities (running, jumping, and playing with a ball) [22,23,24,28], running on a treadmill [25,27,28,29,30,31,32], and cycling on a stationary bike [26]. Seven studies provide data about fatmax (34–54.0% of VO_2_max) and MFO (0.15–0.43 g∙min^−1^) at baseline [25,26,27,28,29,30,32]. All studies reported sustaining heart rate at fatmax during exercise sessions and controlling exercise intensity with a heart rate monitor. In addition, exercise sessions were supervised by a qualified physical exercise instructor. 

Although no studies under dietary treatments were included, eight studies analyzed the diet composition of the participants by nutritional records. In these studies, no changes in diet or caloric intake were found [22,23,24,27,28,29,30,32].

All studies reported the effect of FMT on FM (n = 181); ten of them on BW (n = 175) [22,23,24,25,26,27,28,29,30,32], eight on FFM (n = 135) [22,25,27,28,29,30,31,32], six reported waist circumference change (n = 90) [21,22,23,26,28,31], five on VO_2_max (n = 95) [25,27,28,30,32], and four on MFO [22,23,27,28]. Only two studies evaluated fatmax after FMT [27,28]. 

### 3.6. Effect of FMT on Body Composition

Bodyweight decreased between 1.88–6.52% from baseline values (MD: −4.30 kg; 95% CI: −5.51, −3.10; *p* < 0.01, Figure 3). Change percentage for FM ranged between 5.2–28.09% (MD: −4.03 kg; 95% CI: −4.89, −3.18 kg; *p* < 0.01, Figure 4). Waist circumference reduced between 0.8–5.1% (MD: −3.34 cm; 95% CI: −5.77, −0.91; *p* < 0.01, Figure 5) while FFM remained unchanged (MD = 0.08 kg, *p* = 0.85). The studies were homogeneous (I^2^ = 0%). The sensitivity analysis did not show effect size modifications for BW and FM, but high consistency among the studies (Figure A1). 

### 3.7. Effect of FMT on VO_2_max

The VO_2_max increased between 6.5–12.7% (MD = 2.96 mL∙kg^−1^∙min^−1^, 95% CI: 2.01–3.90 mL∙kg^−1^∙min^−1^, *p* < 0.01, Figure 6). There was no heterogeneity between studies (I^2^ = 0%). 

### 3.8. Publication Bias

Funnel plot analysis showed asymmetrical data distribution only for BW, FM, and VO_2_max, with low sample and short duration studies reporting the lower BW and FM changes. When adjusting effect size by including missing studies values (Figure A2 and Figure A3), effect size increased for BW (0.3 kg, *p* = 0.72) and FM (0.32 kg, *p* = 0.59) pooled effect size, while it was reduced for VO_2_max pooled mean difference (−0.10 mL∙kg^−1^∙min^−1^, *p* = 0.89). No significant publication bias was observed for any variable. 

## 4. Discussion

### 4.1. Effect of FMT on Body Composition

The present meta-analysis demonstrates that FMT is an effective strategy to improve body composition and cardiorespiratory fitness in obese individuals with low physical activity. 

To our knowledge, this is the first meta-analysis of randomized clinical trials that report the positive effect of FMT on FM, WC, and VO_2_max in individuals with obesity, and reveals a considerable reduction of BW (MD: −4.30 kg; *p* < 0.01), with a low heterogeneity across studies (I^2^ = 0%). Similarly, Romain et al., 2012 [15], in their meta-analysis, reported a reduction of BW (−1.95 kg, 95% CI: −3.28, −0.62, *p* < 0.01) by FMT across different populations with low physical activity, including individuals with obesity. Nonetheless, the utility of these results is limited due to:A high heterogeneity among their studies (I^2^ = 85%)Mixing of randomized and non-randomized trials into the analysisMixing of exercise + hypocaloric diet and only exercise treatmentsThey did not evaluate the risk of bias of the studiesOnly one study had a duration > 3 months

The larger effect of FMT on reducing BW could be explained by the comparable intervention period (8–20 weeks), and similar experimental design (randomized clinical trials) among the studies.

An added value of this work is the high reduction of total and abdominal fat after a medium-term FMT (MD: −4.03 kg, *p* < 0.01), which is clinically relevant for obese individuals because adiposity loss reduces metabolic disease and cardiovascular risk [3]. 

The effect size of FMT over FM is robust, as confirmed by the sensitivity analysis (Figure A1), where one-by-one exclusion of each study did not significantly alter pooled effect size nor homogeneity. Likewise, removal of male or adolescent individuals did not affect pooled effect size. Nonetheless, more studies in these populations are needed, in order to analyze how age and sex could affect the response to FMT. In addition, the studies should utilize a specific physical activity for FMT, as fat oxidation rate might vary across different playground activities and heart rate value may not remain stable during the entire exercise session [35].

The present study shows the beneficial medium-term effects of FMT (8–20 weeks) on BW, FM, and VO_2_max, but longer interventions could have better results. Through database search, only one long-term study (two years) about the effect of FMT on BW and body composition was found [36]. The authors report a decrease of 5.1 ± 1.26 kg on BW in patients with obesity (*p* < 0.001), which is higher than the estimated effect size in the present meta-analysis. Nevertheless, this study was not a randomized trial, and the authors did not mention the exercise modality. Besides, males and females were analyzed as a single statistical group, and they did not effectuate a periodical assessment of fatmax to adjust exercise intensity. Therefore, further long-term randomized controlled trials are needed to prove the effect of FMT on body composition and CRF in people with obesity.

As above mentioned, mean fatmax values in subjects with obesity fall within a moderate-exercise intensity domain; however, moderate-intensity training should not be considered as FMT due to large intra-individual variations in fatmax. Evidence for this argument is revealed by the contrast between results of this meta-analysis and those reported by Thorogood et al., 2011 [37], who conclude that six to twelve months (120–140 min per week) of various modes of aerobic exercise at moderate intensity is not an effective weight-loss therapy for patients with obesity. In that meta-analysis of randomized controlled trials, they report a modest BW reduction (MD = −1.7 kg, 95% CI: −2.29 to −1.11). This dissimilarity between Thorogood et al. and the present meta-analyze could be explained by differences in exercise intensity (fatmax vs. moderate) as fatmax exercise results in higher fat utilization. Thereby, FMT might replace moderate-exercise intensity in the classical recommendations for BW management. Nevertheless, additional meta-analyses are necessary to compare the effects of FMT vs. moderate and high-intensity exercise, thereby recommending the best training modality for FM reduction in O/Ob patients.

Concerning FFM, several studies have found skeletal muscle hypertrophy (>7%) after acute and chronic aerobic exercise training [38]. Even as muscle mass accounts for 20% of resting metabolic rate [39], an increase in muscle mass would augment resting energy expenditure. However, the present analysis shows that FFM remains unchanged after FMT. The low physical effort performed during this exercise type is the principal reason to explain the reduced effect of FMT on muscle mass development, as hypertrophy requires high muscle power during skeletal contractions [38]. Pooled effect size in the present meta-analysis denotes the importance of adding resistance exercises at FMT to increase muscle mass and lipid caloric expenditure. 

### 4.2. Mechanisms for Body Fat Reduction

The present meta-analysis demonstrates that FMT is an effective therapy for BF loss; nevertheless, we do not have enough information about the physiological and molecular mechanism that promotes this effect. BF reduction requires to accomplish a negative BF balance; exercise at fatmax guaranties the highest fat utilization during exercise on individuals with obesity. However, maximal fat oxidation in this meta-analysis ranged from low to moderate values (0.15–0.43 g∙min^−1^); whereby, according to the reported exercise duration (120–360 min per week), the total fat oxidation (8.25–25.8 g) would not be enough to achieve a negative fat balance in 24 h. 

To our knowledge, only one study has reported the effect of FMT on resting fat oxidation in a fasted state [13]. In that study, ten weeks of FMT did not increase resting fat oxidation in obese women: 0.08 ± 0.09 vs. 0.07 ± 0.03 g∙min^−1^ for pre and post-intervention measures. Besides, Lazzer et al., (2010) [40] reported that a 45-min of fatmax did not increase fat oxidation in severely obese adolescents, as basal values were reached 15 min after exercise ceased. Based on these results, it appears that fatmax does not increase resting fat oxidation in people with obesity; however, future metabolic studies should evaluate physical, physiologic, and biochemistry parameters in order to clarify how FMT reduces BW and FM.

Exercise training increases fatty acid oxidation in skeletal muscle due to the increment of muscle mitochondrial biogenesis and sarcolemmal transport proteins via the PGC 1-α activation pathway [5,41]. Since 2001, Jeukendrup and Achten proposed that FMT could enhance lipid metabolism in skeletal muscle as lipid metabolic pathways are maximally activated at fatmax [9]. Nevertheless, we do not get enough information about the effect of FMT on molecular markers of lipid metabolism. In this sense, none of the here included studies evaluated the impact of FMT on 24 h fat oxidation, but four of them reported a significant increase in MFO (∆: 0.03–0.07 g∙min^−1^) after 8 to 10 weeks of FMT [22,23,27,28]. Data from Bordenave et al., 2008 and Tan et al., 2016 demonstrate an increase in skeletal muscle mitochondrial respiratory capacity, coupling with an augmented lipoprotein lipase activity after 10 weeks of FMT in overweight and obese individuals [14,27], important benefits in those obese subjects with an impaired ability to use fatty acids as energy substrate. However, more research is needed to evaluate the effect of FMT on the activity and expression of relevant proteins that regulate lipid metabolisms, like PGC1-α, hormone-sensitive lipase, fatty acid translocase, carnitine palmitoyltransferase 1 and 2, and 3-hydroxy acyl-CoA dehydrogenase.

In addition to lipid metabolism, data from Tan et al. (2018) [29] and Jiang et al. (2020) [32] demonstrate that FMT could improve glucose homeostasis. According to these studies, 12–16 weeks of running at FMT reduces fasting blood glucose, increases blood adiponectin, and reduces insulin resistance in sedentary obese patients with T2D. The binding of adiponectin to AdipoR1 receptor in skeletal muscle results in phosphorylation of AMPK, increasing mobilization of glucose transporters (GLUTs) to the cell membrane, and the glucose uptake for future use [42]. Although these findings indicate a positive effect of FMT on diabetic individuals’ health, future studies should determine the effect of FMT on blood and muscle carbohydrate markers like glucose, insulin, Leptin, Ghrelin, AMPK, GLUTS, among others.

### 4.3. Effect of FMT on Cardiorespiratory Fitness

It is well supported that moderate and high-intensity exercise improves various health parameters, including increase in VO_2_max, stroke volume, arteriovenous O_2_ difference, and skeletal muscle mitochondrial biogenesis [43,44,45]. All of them improve the ability to perform work and delay fatigue. In the present review, we found a slight VO_2_max increment (MD = 2.96 mL∙kg^−1^∙min^−1^, 95% CI: 2.01–3.90 mL∙kg^−1^∙min^−1^) after eight to sixteen weeks of FMT. This augment is explained by the increase of absolute oxygen uptake (0.05–0.16 L∙min^−1^) and not by body weight reduction only. The observed change represents a mean increase of 10% in people’s CRF and is similar to those reported by Murphy et al. (2.73 mL∙kg^−1^∙min^−1^) [46]; such increase is associated with a decrease in diastolic blood pressure. In addition, Wang et al. found that 10% increase of CRF after 10 weeks of FMT is enough to reduce systolic blood pressure (~13 mmHg) and improve stroke volume (~5 mL) and ejection fraction (~4.6%) in poor fitness individuals [13]. Based on these data, FMT seems to improve CRF and reduce cardiovascular risk in obese patients, at least during the early stages of physical conditioning.

### 4.4. Weakness in the Analyzed Studies

(I) The articles here incorporated do not explain whether or not the people studied were in energy balance. They only report that there were no changes in energy intake or macronutrient composition in diet during the studies. (II) Lipid oxidation was not measured during the prolonged exercise test or training sessions. (III) Researchers did not periodically evaluate fatmax. (IV) The authors did not specify if participants were in the postprandial or post-absorptive state during the tests or exercise sessions. (V) Studies did not report exercise energy expenditure. 

Recently, Özdemir et al. (2019) reported that MFO determined by an incremental treadmill test in sedentary lean individuals is not sustained during 40 min of walking at fatmax [47]. Exponential decay is observed in fat oxidation rate and a plateau after 16 min, in spite of the heart rate at fatmax being kept constant. Unpublished data in our laboratory show similar results during the first 15 min of exercise at fatmax, with a return to the programmed fat oxidation rate after 60 min (Figure A4). Therefore, exercises less than 60 min at fatmax may not correspond to the theoretical fat oxidation rates. 

Moreover, exercise training improves CRF, which is positively associated with fatmax and MFO [7,17]. The present work shows that FMT significantly increases VO_2_max; however, none of the included studies re-evaluate fatmax after several weeks of training to adjust exercise intensity if needed. Data from Tan et al., 2016 [27,28] suggest that more than eight weeks of FMT are need in order to increase fatmax; however, future studies should periodically evaluate MFO and fatmax. Furthermore, specifying fasting status before the exercise test is essential, mainly because the carbohydrate intake increases blood insulin, increasing carbohydrate oxidation, and decreasing lipid oxidation [48]. In addition, studies should report information about the total energy expenditure during exercise, necessary to analyze the impact of FMT on body mass loss. 

### 4.5. Strengths

(I) This work performs a systematic search in four preeminent databases that group most of the scientific journals: PubMed, SCOPUS, EBSCOhost, and ScienceDirect. (II) Manuscript search was carried out using different synonym terms to define fatmax, thereby reducing information loss. (III) The risk of bias was assessed by two independent reviewers, following the guidelines of the Cochrane handbook of systematic reviews. (IV) The effect size for BW, FM, and VO_2_max shows a high statistic power for our meta-analysis (Z > 6.0, *p* < 0.001) and high homogeneity (I^2^ = 0). (V) The sensitivity analysis for BW and FM outcomes shows high robustness. (IV) No significant publication bias was observed for any of the analyzed variables. In fact, the analysis suggests an increase in BW and FM effect size if studies with low sample size and ≥10 weeks of duration were available in the literature. 

### 4.6. Limitations

(I) The present review focused on peer-reviewed articles written in English and Spanish, excluding possible relevant data from grey literature and other languages. Through database search, we identified three articles about FMT written in French; however, due to the authors’ language limitations, these documents could not be assessed for eligibility. (II) Only 11 manuscripts were found; this merits an update in the short- and medium-term.

## 5. Conclusions

Eight to twenty weeks of FMT, with 120–360 min per week of training volume, reduces body weight, BF, and improves cardiovascular fitness in low physical activity individuals with obesity. The molecular mechanisms that explain these changes are mostly unknown.

## Figures and Tables

**Figure 1 ijerph-17-07888-f001:**
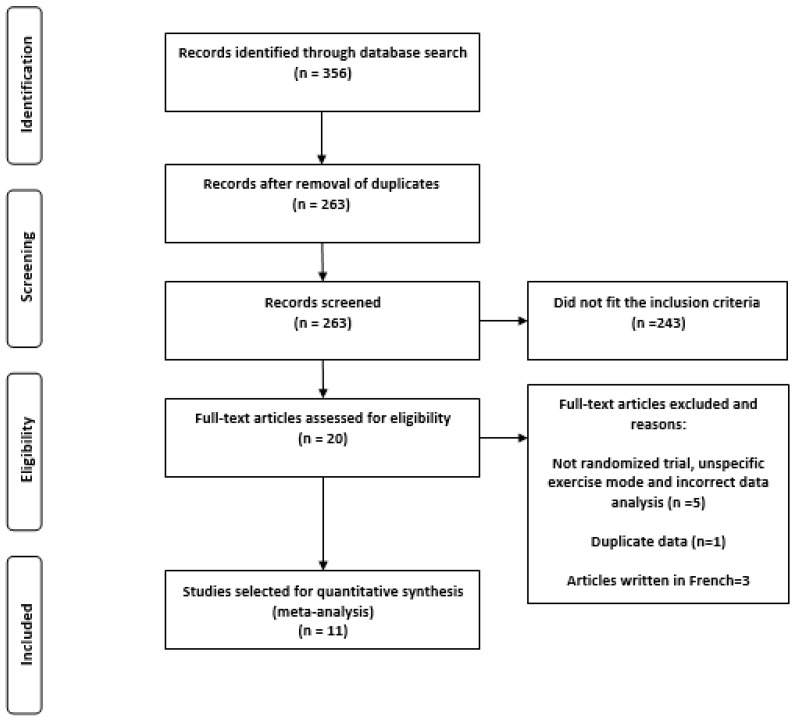
Flow chart of selected articles.

**Figure 2 ijerph-17-07888-f002:**
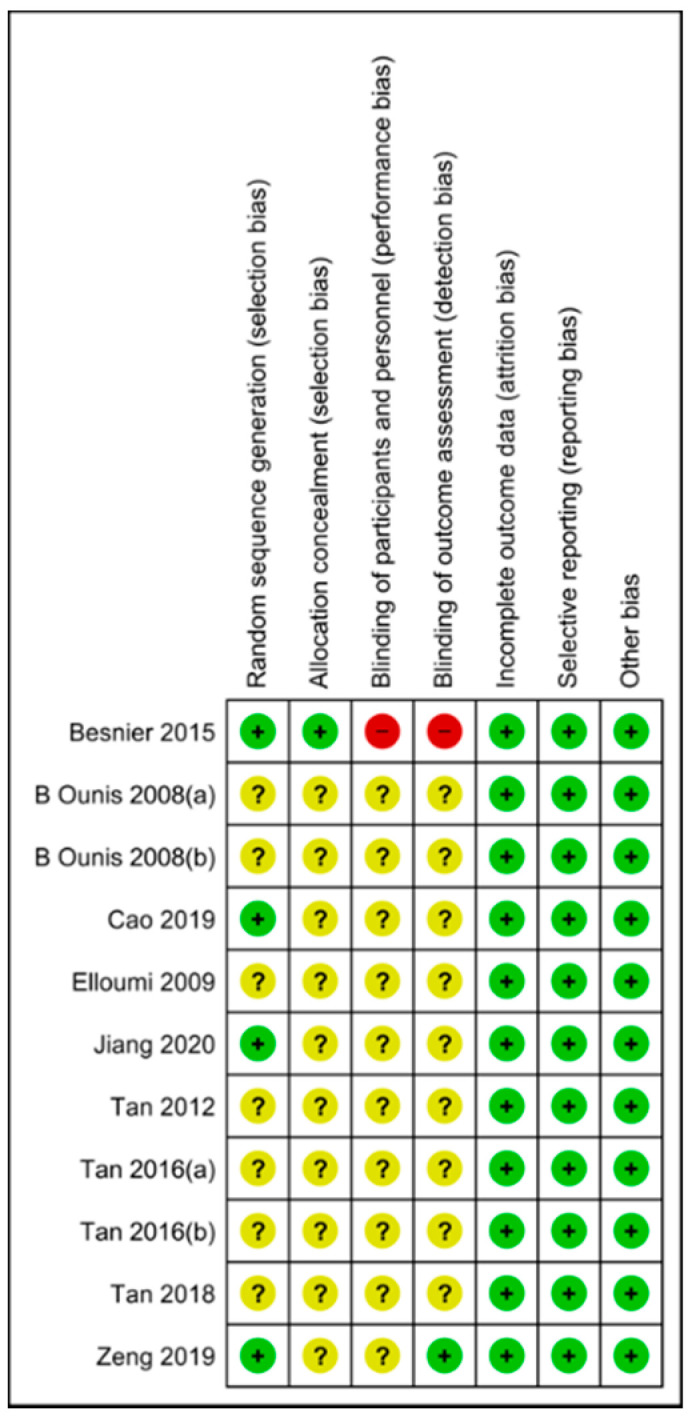
Risk of bias summary. Authors’ judgements about each risk of bias item for every included study. Green means low risk of bias; yellow, unclear risk of bias and red, high risk of bias.

**Figure 3 ijerph-17-07888-f003:**
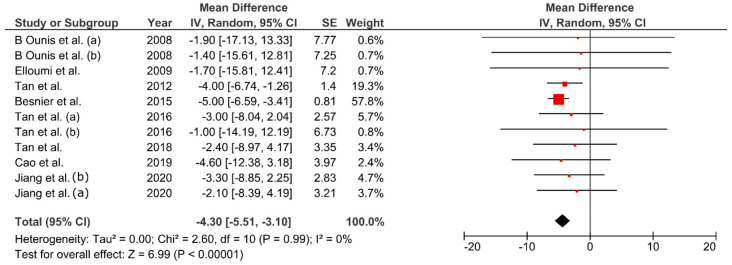
Pooled mean difference and 95% confidence interval for body weight (kg) change in fatmax training experimental groups: baseline vs. after intervention. Negative values indicate an improvement in body weight.

**Figure 4 ijerph-17-07888-f004:**
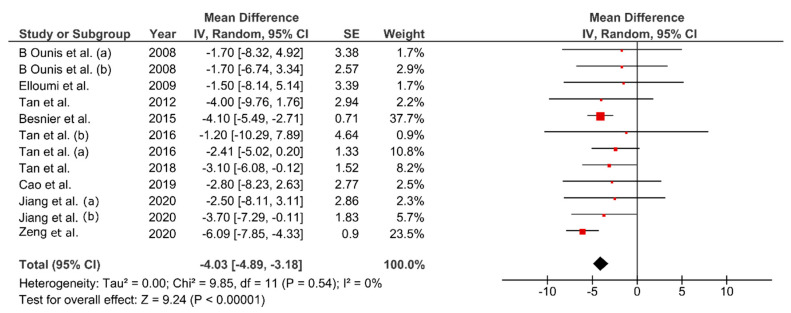
Pooled mean difference and 95% confidence interval for fat mass (kg) change in fatmax training experimental groups: baseline vs. after intervention. Negative values indicate an improvement in body composition.

**Figure 5 ijerph-17-07888-f005:**
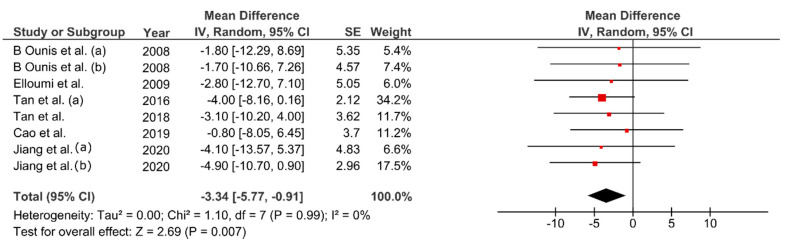
Pooled mean difference and 95% confidence interval for waist circumference (cm) change in fatmax training experimental groups: baseline vs. after intervention. Negative values indicate an improvement in body composition.

**Figure 6 ijerph-17-07888-f006:**
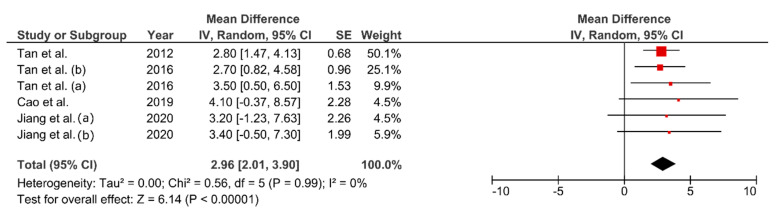
Pooled mean difference and 95% confidence interval for maximal oxygen consumption (ml∙kg^−1^∙min^−1^) change in fatmax training experimental groups: baseline vs. after intervention. Positive values indicate an improvement in cardiorespiratory fitness.

**Table 1 ijerph-17-07888-t001:** Descriptive characteristics of the included studies.

Study	Participants, Age (year)	BMI (kg∙m^−2^)	Exercise Protocol	∆ BW (kg)	∆ FM (kg)	∆ Vo_2_max (mL∙kg^−1^∙min^−1^)
Ounis et al., (2008)a [22]	8 boys (13.1 ± 0.1)	30.2 ± 4.2	FMT: playground activities; 4 day/week; 90 min/session; 8 weeks	−1.90 ± 15.54	−1.70 ± 6.76	NR
Ounis et al., (2008)b [23]	6 girls (13.1 ± 0.1)	30.6 ± 2.3	FMT: playground activities; 4 day/week; 90 min/session; 8 weeks	−1.40 ± 12.55	−1.70 ± 4.45	NR
Elloumi et al., (2009) [24]	7 boys (13.1 ± 0.7)	30.3 ± 3.2	FMT: playground activities; 4 day/week; 90 min/session; 8 weeks	−1.70 ± 13.46	−1.50 ± 6.34	NR
Tan et al. (2012) [25]	29 women (20–23)	27.5 ± 1.9	FMT: running; 3 day/week; 60 min/session; 8 weeks	−4.00 ± 5.33 *	−4.00 ± 11.19 *	2.80 ± 2.58 *
Besnier et al., (2015) [26]	33 women (30.5 ± 5.9)	33.3 ± 3.8	FMT: cycling; 4 day/week; 55 min/session; 20 weeks	−5.00 ± 3.29 *	−4.10 ± 2.88 *	NR
Tan et al., (2016)a [27]	15 women (50.7 ± 5.5)	28.5 ± 2.1	FMT: running; 5 day/week; 60 min/session; 12 weeks	−3.00 ± 7.03 *	−2.41 ± 3.64 *	3.50 ± 4.19 *
Tan et al., (2016)b [28]	11 boys (9.0 ± 1.0)	27.1 ± 4.3	FMT: playground activities; 5 day/week; 60 min/session; 10 weeks	−1.00 ± 15.78 *	−1.20 ± 10.88 *	2.7 ± 2.25 *
Tan et al. (2018) [29]	16 elderly women with T2D (63.0 ± 2.3)	26.6 ± 3.1	FMT: running; 3 day/week; 60 min/session; 12 weeks	−2.4 ± 9.47 *	−3.10 ± 4.29 *	NR
Cao et al. (2019) [30]	13 women (63.8 ± 5.9)	28.0 ± 2.9	FMT: running; 3 day/week; 60 min/session; 12 weeks	−4.60 ± 10.12 *	−2.80 ± 5.78 *	4.1 ± 5.81 *
Zeng et al. (2019) [31]	18 women (21.1 ± 1.6)	26.6 ± 3.1	FMT: running; 3 day/week; 45 min/session; 12 weeks	NR	−6.09 ± 2.7 *	NR
Jiang et al. (2020) [32]	13 elderly women with T2D ^a^ (63.9 ± 6.1)	26.6 ± 2.2	FMT: running; 3 day/week; 60 min/session; 16 weeks	−2.10 ± 8.18 *	−2.50 ± 7.29 *	3.4 ± 5.07 *
	14 elderly men with T2D ^b^ (63.9 ± 6.1)	26.9 ± 2.1	FMT: running; 3 day/week; 60 min/session; 16 weeks	−3.30 ± 7.84 *	−3.7 ± 4.84 *	3.2 ± 5.97 *

Data represents mean ± SD, mean difference between baseline and post-intervention measurements (∆). FMT, exercise training at maximal fat oxidation; BW, body weight; FM, fat mass; VO_2_max, maximal oxygen consumption; T2D, type 2 diabetes mellitus. NR, data not reported. Note: recreational activities include walking, running, and playing with a ball. * *p* < 0.05 between the pre and posttest within training groups.

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
