# Peer review of "Chronic Effect of Fatmax Training on Body Weight, Fat Mass, and Cardiorespiratory Fitness in Obese Subjects: A Meta-Analysis of Randomized Clinical Trials"

_ijerph, 2020, doi:10.3390/ijerph17217888_

Round 1

Reviewer 1 Report

The article has been improved; however, there are answers that I do not agree with or have been omitted. Please check them

Line 62. For the knowledge of the authors.... Likewise, don’t use 1stperson, always 3er person.

R: The vast majority of the document is writing in the third person; however, the indiscriminate use of the third person is a literary and grammatical error.

Reviewer: Are you sure? I never ear that. Please change for 3er person.

Table 1. Δshould not carry standard deviation? And participants’ age or BMI? I think too 2

R: Mean difference(Δ)and the standard error of the difference between baseline and post-treatment measures are provided in the graphs.The authorsconsider that including the standard deviation of the difference in the table would be repetitive.Standard deviation for age and BMI was added to the table.

Reviewer: Please include Standard deviation of Δ BW (kg), Δ FM (kg) and Δ VO2max (ml∙kg−1∙min−1) in table 1

6.Line 110. Why were considered Children and adolescents as obese if their BMI value was 110 abovethe 95th percentile? Include reference

Reviewer: This comment has not been answered

8.Include funnel plots to visual interpretation to confirm or refute publication bias.

R: Publication bias analysis was incorporated to statistical analysis (2.7), results (3.8), and strengths (4.7) sections.According to Cohens Criteria, input of missing studies results on a moderate increase of BW effect size (0.62), large increase of FM effect size (0.95) and low reduction of VO2 (0.26).

Reviewer: Please include funnel plots for visual interpretation

Author Response

Reviewer 2

Comments and Suggestions for Authors

The article has been improved; however, there are answers that I do not agree with or have been omitted. Please check them

Line 62. For the knowledge of the authors.... Likewise, don’t use 1stperson, always 3er person.

R: Done. The manuscript is writing in the third person. Likewise, was replaced by “besides” and the word “this” was changed to “that”. Following the reviewer suggestion (line 62)

Table 1. Δ should not carry a standard deviation? And participants’ age or BMI? I think too 2

R: Done. Mean difference (Δ) and the standard error of the difference between baseline and post-treatment measures are provided in the graphs.

Reviewer: Please include the Standard deviation of Δ BW (kg), Δ FM (kg), and Δ VO2max (ml∙kg−1∙min−1) in table 1

R: Done. Mean difference standard deviations were added for Δ BW (kg), Δ FM (kg), and Δ VO2max (ml∙kg−1∙min−1) in Table 1.

6. Line 110. Why were considered Children and adolescents as obese if their BMI value was 110 above the 95th percentile? Include reference

Reviewer: This comment has not been answered

R: Done. Reference was added to de text and references section (lines 179 and 525).

8. Include funnel plots to visual interpretation to confirm or refute publication bias.

R: Done. Publication bias analysis was incorporated into statistical analysis (2.7), results (3.8), and strengths (4.7) sections. According to the Cohens Criteria, the input of missing studies results in a moderate increase of BW effect size (0.62), a large increase of FM effect size (0.95), and a low reduction of VO2 (0.26).

Reviewer: Please include funnel plots for visual interpretation

R: Done. Funnel plots were included in figure 9. As can be seen, asymmetrical data distribution was founded for BW, FM, and VO2max outcomes but not for FFM and WC. According to the Cohens Criteria, the input of missing studies results in a moderate increase of BW effect size (0.62), a large increase of FM effect size (0.95), and a low reduction of VO2 (0.26). Nevertheless, effect size modifications were not statistically significant (Section 3.8).

Reviewer 2 Report

The reviewer thanks the authors for making several revisions that improve this manuscript. Specifically, the inclusion of waist circumference data strengthens the author's conclusions and further supports the primary hypothesis.

One minor clarification is included below.

Line 218: The authors have not yet provided evidence that these individuals have ‘poor physical fitness’. Without such data, the authors may not be positioned to make this statement.

R: This statement is based on body composition and physical activity(lines 180-183)reported for individuals at baseline.

Now line 243: To clarify, the important distinction I would like to make for the context of this comment is that physical activity habits and cardiorespiratory fitness represent different variables. That is, an individual performing high amounts of physical activity does not necessarily have high cardiorespiratory fitness (as assessed via oxygen uptake), and the opposite is also true. Thus, the authors would be encouraged to consider changing "poor physical fitness" to "low physical activity". Alternatively, because not all manuscripts reported physical activity habits (lines 182-185), another approach could be to simply remove "poor physical fitness".

Author Response

R: Thanks. The term “poor physical fitness” was changed to “low physical activity” throughout the entire manuscript (lines 245, 250, and 394), in accordance with the reviewer's recommendation.

Round 2

Reviewer 1 Report

Perfect. Congratulations.

This manuscript is a resubmission of an earlier submission. The following is a list of the peer review reports and author responses from that submission.

Round 1

Reviewer 1 Report

Reviewer comments have been addressed.

Reviewer 2 Report

Abstract.

Line 17. Delete Therefore. Change performed

Line 27. Include means of MD

Introduction

Line 62. For the knowledge of the authors…. Likewise, don’t use 1st person, always 3er person.

I think the reason for the realization of this work is missing. Why is it important to know the Fmax in obese patients? Does it have any clinical value? Review objective justification

Materials and Methods

Line 76: Include authors initials

Line 78. Include the search’s equation no only keywords.

Line 80. Why form January 01, 2001 and not form January 01, 1990 for example?

Figure 1. Use Original PRIMA Flow chart

Line 85. Why don’t use Portuguese or German for example?  I think removing some languages might stop including some interesting articles. If the authors have already reviewed that there are no articles in these languages, include them within the inclusion criteria.

Table 1. Δ should not carry standard deviation? And participants’ age or BMI? I think too

Line 110. Why were considered Children and adolescents as obese if their BMI value was 110 above the 95th percentile? Include reference

Line 112. Include references or data for know what cutoffs-values you have used

Lines 140-142. Indicate the number of the manuscript that you delete for each exclusion criteria

Include funnel plots to visual interpretation to confirm or refute publication bias.

Figure 3, 4 and 5. Include some information about if is better go to left (-10, -5) or right (2, 10).

Discussion

The discussion is well oriented and discussed. Same for the conclusion

Reviewer 3 Report

General comments:

The authors state that the primary purpose of this article is to define the effect of FMT on body composition, but only one crude measure of body composition (BMI) is included in the article. The authors would be encouraged to include a combination of waist circumference, hip circumference, waist to hip ratio, and/or body fat changes with FMT. Without such data or comparisons, it is difficult for the reader to fully interpret the results of fatmax training on body composition.

Within the context of weight loss studies, it is important to mention how changes in absolute VO2 and VO2 relative to fat-free mass were changed. This allows for the separation of effects do to weight loss alone or true increases in oxygen utilization capacity. Without such data or comparisons, it is difficult for the reader to fully interpret the results of fatmax training on cardiorespiratory fitness.

A major limitation that should be mentioned is that this article considered the effects of fatmax training on body mass and fat mass loss. However, with the goal for prescribing exercise, it would be much more insightful to compare fatmax training versus moderate- or high-intensity exercise training. In short, it is helpful to know that fatmax training reduces body weight, but if it is less effective than other training modalities that would be the more important question to consider. Thus, the authors are encouraged to list this point as a limitation.

Specific comments:

Line 34: is “and” needed?

Lines 52-53: is the following clearer? “reduced fat mass by 2 kg, augmented VO2max by 10-20%...”

Lines 53 and throughout: VO2max is a rate, so there should be a dot over the ‘V’

Line 58: ‘in’ instead of ‘on’?

Line 72-73: ‘on’ body composition?

Line 151: ‘no’ dropouts?

Line 154: ‘no’ other bias?

Line 181: ‘interval’ training?

Line 188: ‘heart’ rate?

Line 189: is “also” a better fit than “besides”?

Lines 198-198: It may be redundant to write both ‘decrease’ and include minus symbols prior to values (i.e., bodyweight decreased between 1.4 and 5.0 kg). Also, it is atypical to include the same data (e.g., -4.3 kg weight loss and 4.03 kg fat mass loss) in both the text and figures. The authors should consider only reporting these raw (plotted) and numerical data in figures 3 and 4.

Table 1, lines 198-200, Figures 3 and 4: It would be helpful for readers if absolute values were supplemented with percent changes, especially when adolescents and adults are grouped together to come up with mean values reported (e.g., 4.3 kg body weight loss).

Lines 210-22: it is atypical to include the same data in both the text and figures. The authors should consider only reporting these raw (plotted) and numerical data in figure 5.

Line 218: The authors have not yet provided evidence that these individuals have ‘poor physical fitness’. Without such data, the authors may not be positioned to make this statement.

Line 229: This seems to be unclear as written. The authors are encouraged to consider an edit similar to the following ‘The larger effect of FMT on reducing bodyweight could be explained…’

Line 284: ‘seem that fatmax…’? Also, should there be an edit to ‘individuals with obesity’ rather than ‘obese people’?

Line 285: Would ‘how’ be a better fit than ‘why’. It seems overly obvious that with caloric intake equal any bout of exercise (fatmax or not) completed over several weeks would reduce bodyweight.

Line 292-293: ‘on 24 h fat oxidation’?

Lines 295-297: This statement is unclear as written. What about ‘increase in skeletal muscle mitochondrial respiratory capacity, coupled with an augmented lipoprotein…’?

Lines 297-298: It is unclear what the sentence starting with ‘Valuable’ is in reference to.

Line 318: It doesn’t appear that the term ‘besides’ is used correctly here. Perhaps ‘also’ would be a better fit.

Figure 7: While the reviewer appreciates the relevance of figure 7, there is no mention of an IRB approval to collect and publish such data. Nor is there mention of the methods used (i.e., was a chest heart strap and indirect calorimetry used? Also, there was no mention of these data in the results section.

Lines 335-336: It does not seem sufficient to make a concluding statement (e.g., “Therefore, exercise less….”) from data obtained in a single 60-minute bout of exercise. The authors will need to provide citations to keep such a statement included.

Lines 337: Is exercise training or CRF positively associated with fatmax?

Figure 6: Is ‘kg’ the correct unit for effect size?